# Prevalence and predictors of depression, anxiety, and stress among adults in Ghana: A community-based cross-sectional study

Hubert Amu[1]*, Eric Osei[1,2], Philip Kofie[1], Richard Owusu[1], Samuel Adolf Bosoka[1], Kennedy Diema Konlan[2,3], Eunji Kim[4], Verner Ndudiri Orish[5], Raymond Saa-Eru Maalman[5], Emmanuel Manu[1], Phyllis Atta Parbey[1], Farrukh Ishaque Saah[1], Hadiru Mumuni[3], Prince Kubi Appiah[1,6], Joyce Komesuor[1], Martin Amogre Ayanore[1], Gregory Kofi Amenuvegbe[1], Siwoo Kim[6], Hajun Jung[6], Martin Adjuik[1], Elvis Enowbeyang Tarkang[1], Robert Kaba Alhassan[7], Ernestina Safoa Donkor[3], Francis Bruno Zottor[1], Margaret Kweku[1], Paul Amuna[1], So Yoo Kim[6], John Owusu Gyapong[8]

1 School of Public Health, University of Health and Allied Sciences, Hohoe, Ghana, 2 Department of Public Health, Yonsei University Graduate School, Seoul, Republic of Korea, 3 School of Nursing and Midwifery, University of Health and Allied Sciences, Ho, Ghana, 4 Korean Foundation for International Healthcare, Seoul, Korea, 5 School of Medicine, University of Health and Allied Sciences, Ho, Ghana, 6 Asian Institute for Bioethics and Health Law, College of Medicine, Yonsei University, Seoul, Korea, 7 Directorate of International Affairs, University of Health and Allied Sciences, Ho, Ghana, 8 Office of the Vice-Chancellor, University of Health and Allied Sciences, Ho, Ghana

* hamu@uhas.edu.gh

**Data Availability Statement:** The data has been made available in the Supporting information files.

## Abstract

### Introduction

Over the past two decades, there have been several global interventions including the Sustainable Development Goals (SDGs), aimed at improving health outcomes. Despite efforts by countries to achieve the SDG targets, mental health challenges remain major public health concerns globally. We examined the prevalence and predictors of depression, anxiety, and stress as well as the comorbidities of these mental health issues among adults.

### Materials and methods

This was a community-based cross-sectional study conducted among 2456 adults in four districts of the Volta Region of Ghana using data from the UHAS-Yonsei University Partnership Project. We analysed the data using frequency, percentage, mean, standard deviation, correlation, and binary logistic regression.

### Results

Overall, 51.8% of the participants had at least one of the mental health issues examined. The prevalence of a mental health issue was 25.2%, 53.3%, and 9.7% for depression, anxiety, and stress respectively. Participants constituting 8.3% experienced all three mental health issues as comorbidities. Participants' level of formal education and income significantly predicted depression, anxiety, and stress respectively at the multivariable level. Adults with a tertiary level of education were, for instance, 68% (AOR = 0.32, 95%CI = 0.15–

**Funding:** The study is part of an ongoing project funded by the UHAS-Yonsei University Partnership Project. The funders had no role in study design, data collection and analysis, decision to publish, or preparation of the manuscript.

**Competing interests:** The authors have declared that no competing interests exist.

0.66), 65% (AOR = 0.35, 95%CI = 0.17–0.73), and 50% (AOR = 0.50, 95%CI = 0.33–0.76) less likely to experience depression, anxiety, and stress, respectively compared with those who had no formal education.

## Conclusion

The majority of our study participants either experienced depression, anxiety, or stress. There were quite high comorbidities of the mental health issues among the adult population. To accelerate progress towards the achievement of SDG 3.4 target of promoting mental health and wellbeing for all by the year 2030, there is a need for effective implementation of the country's 2012 Mental Health Act which makes provisions for the establishment of a Mental Health Fund. This could improve the financial circumstances of indigenes as income has been realised in the present study as an important factor influencing depression, anxiety, and stress among the adult population.

## Introduction

Over the past two decades, there have been several global interventions to improve health outcomes. The Sustainable Development Goals (SDGs) which were promulgated in the year 2015, for instance, seek to ensure that by the year 2030, developing countries across the globe can achieve 17 development-oriented goals [1]. Goal three seeks to ensure healthy lives and promote well-being for all at all ages and target 3.4 specifically aims at promoting mental health and well-being for all by the year 2030 [1]. Despite efforts by the various developing countries to achieve the SDG targets, mental health issues remain major public health concerns as about 792 million (10.7%) people experiencing such issues, placing them among the leading causes of disability worldwide [2].

Depression is one of the main mental health conditions responsible for the global burden of disability with an estimated 264 million people experiencing it [2,3]. Depression is characterised by sadness, loss of interest or pleasure, feelings of guilt or low self-worth, disturbed sleep or appetite, tiredness, and poor concentration [4,5]. These characteristics greatly affect the quality of life and overall well-being of the individuals who experience the condition. Anxiety disorder is also a mental health condition of public health concern. Anxiety disorders are a group of mental disorders characterised by feelings of anxiety and fear, and they include generalised anxiety disorder, panic disorder, phobias, and social anxiety disorder, obsessive-compulsive disorder, and post-traumatic stress disorder. Anxiety disorders are the most common mental health conditions worldwide as one in 13 people globally live with it [6].

In addition, stress is a major mental state defined as "the non-specific response of the body to any demand for change" [7,8]. It has been named by the World Health Organization (WHO) as the Health Epidemic of the 21st century [9,10]. Stress adversely influences the world economy as a section of the world human resource go through some form of stress, hereby, limiting the per capita income of nations [9,10].

Depression, anxiety and stress often interrupt the regular activities of people, such as the inability to work effectively or taking care of the family [11]. Personal neglect is commonly experienced by persons with depression, where the individual least prioritises their hygiene and hence, predisposing them to communicable and non-communicable diseases [6]. It is also not uncommon for an individual with an anxiety disorder to also live with depression or vice

versa as nearly one-half of those diagnosed with depression are also diagnosed with an anxiety disorder [6]. In Sub-Saharan Africa (SSA), depression, anxiety, and stress are major mental health issues of public health concern with depression and anxiety recording 9% and 10% prevalence rates respectively among the general population [12].

In Ghana, a major policy intervention towards improving the mental health of indigenes was the promulgation of the Mental Health Act in the year 2012 [13]. The act admonished the establishment of a community-based mental health system to holistically address the mental health needs of community members. Various studies in Ghana have assessed mental health issues. Kugbey et al. [14], for instance, examined the influence of social support on the levels of depression, anxiety, and stress among students and found high proportion of mild to severe depression and anxiety (57% and 84%, respectively) and 49% had stress. The finding further showed that social support significantly predicted depression and stress while significant differences in all three in terms of sex. Arhin et al. [15] studied anxiety, stress, and depression and bullying victimization among school-going adolescents. They found that psychological distress comprising depression, anxiety, and stress positively predicted bullying victimization among adolescent. Asante and Andoh-Arthur [16] examined the prevalence and determinants of depression among university students and observed that 39.2% had depressive symptoms which was associated with limited social support, religion, heavy alcohol consumption, and traumatizing experiences. Bonsu et al. [17] also assessed the mediation effects of depression and anxiety on social support and quality of life among caregivers of persons with severe burns injury. The authors reported that depression and anxiety negatively impacted the participants' quality of life. Other factors have been identified to influence psychological distress in Ghana and other countries in the sub-region. For instance, among healthcare workers in Western Ghana, age, educational background, marital status, and workload were reported to account for level of stress [18]. Additionally, age, sex, and social support were predictors of depression, anxiety and stress among people living with HIV/AIDS in Ghana [19], occupational stress also predicted depression among Ghanaian telecommunication workers [20], and sex, non-prescription drug use, caffeine consumption, and job prospects have significant impact on depression, anxiety, and stress among waiters in Ghana [21].

There is currently, however, a paucity of community-based empirical research on the prevalence of depression, anxiety, and stress in the country. Aside from being non-community-based and among the general population, the previous studies focused on the prevalence of depression, anxiety, and stress independently, with no emphasis on their co-existence in terms of analyses conducted. This study, therefore, bridges the gap by demonstrating the comorbidity of depression, anxiety, and stress among adults. This study is the first community-based research to show the concurrent prevalence of depression, anxiety, and stress in the adult Ghanaian population and their associated factors. It would, therefore, contribute immensely to the literature on mental health in Ghana and the rest of SSA.

## The UHAS-Yonsei partnership project

We conducted this study as part of a partnership project between the University of Health and Allied Sciences (UHAS) in Ghana, and the Yonsei University in South Korea. The Public Health Educational Capacity Development of UHAS is a 4-year collaborative project which started between the two universities in April 2017 and aims at empowering UHAS to foster outstanding health care professionals who can contribute to the academic development and the national policy advancement of healthcare in Ghana [22]. It is funded by the National Research Foundation of Korea (NRF) [22]. The project specifically seeks to develop the capacity of faculty of UHAS in teaching and research, remodelling the current educational

curriculum, and reorganizing vocational training. Vocational training is the flagship skills acquisition programme of UHAS geared towards enabling students to become properly grounded health professionals by the time they graduate from the university [22]. The School of Public Health (SPH), School of Nursing and Midwifery (SONAM), and School of Medicine (SOM) are the project participating schools out of the six schools in UHAS. To understand the health needs of communities in Ghana, a community health needs assessment survey was conducted under the project [18] from 13th October 2018 to 8th February 2019 in the Volta Region.

## Materials and methods

We adopted the 'Strengthening the Reporting of Observational Studies in Epidemiology' (STROBE) statement in conducting this research and writing the manuscript (See S1 Table).

### Study design and data source

This was a community-based descriptive cross-sectional study using data from the community health needs assessment survey conducted under the UHAS-Yonsei University Partnership Project. There were female, male, and household interviews conducted as part of the survey. The household interviews focused on the size of the households, water and sanitation, fuel for cooking, and source of lighting. The male and female interviews focused on socio-demographic characteristics (age, ethnicity, religion, educational status, main occupation, average monthly income, and place of usual residence), dietary salt, physical activity, history of raised blood pressure, history of raised blood sugar, history of raised total cholesterol, alcohol consumption, malaria history, hepatitis history, depression, anxiety, stress, and use of health facility. Other variables were contraception, sexual behaviour and sexually transmitted infections, antenatal care, place of delivery, post-natal care, breastfeeding and complementary feeding practices, oral health and risk factors. Our study utilised data from the male and female interviews on the socio-demographic characteristics of participants and variables regarding depression, anxiety and stress.

### Study setting and population

The survey was conducted in four districts of the then Volta Region of Ghana. These were the Ho West District, Nkwanta South Municipality (now in the Oti Region), Hohoe Municipality, and Ketu South Municipality. The Volta Region is one of the ten (now 16) administrative regions in Ghana [23]. The region is located between latitudes 50 45"N and 80 45"N along the southern half of the eastern border of Ghana, which it shares with the Republic of Togo. It shares boundaries to the west with Greater Accra, Eastern and Brong Ahafo regions, to the north with the Northern Region, and has the Gulf of Guinea to the south. The region's total land area is 20,570 square kilometres, representing 8.7% of the total land area of Ghana [23].

The Volta Region has a total of 326 health institutions out of which 242 are administered by the Ghana Health Service (GHS), 18 are mission-owned, one is quasi-government, and 65 are privately-owned [23]. Except for Krachi East, Nkwanta North, and Adaklu, every district in the region has a district-level hospital, either government- or mission-owned. The region is divided into 25 administrative Municipal/District Assemblies [23] headed by Municipal/District Chief Executives. The population of the Volta Region based on the 2010 Population and Housing Census figures was 2,491,293 [24]. Out of this, males constituted 1,223,722 while females formed 1,239,279 [24]. The project's survey population comprised men and women aged 15–59 years as well as caregivers of children aged 6 months to 13 years who resided in the

selected districts/municipalities. For the component making up this current publication, however, only adults 18+ years old were included.

## Sampling procedures

A multi-stage cluster sampling procedure was used to select participants from households. A cluster was defined as a collection of households with identifiable geographical boundaries (for instance a town/village). At the first stage, the districts in the region were grouped into three ecological zones (savanna, middle, and coastal zones). Using a simple random sampling procedure, one district was selected from the savanna (Nkwanta South) and Coastal (Ketu South) zones while two were selected from the middle zone (Ho West and Hohoe). The selection of two from the middle zone was because the majority of districts in the region were located in the middle zone. The next stage of the cluster sampling was the selection of communities from each of the four districts. This was then followed by the selection of households in each of the communities. Selection of the households employed the WHO cluster sampling technique [25]. This was done by first locating a geographical landmark (e.g. market, community centre) in each community and spinning a bottle. The first household in the direction to which the mouth of the bottle pointed indicated the starting point of the survey. After this, every other 4th household in that direction was then entered into. This number was chosen based on estimated population's sampling fraction derived. In this study, a household comprised persons living under the same roof who ate together.

According to the 2010 Population and Health Census of Ghana, the total number of households in Hohoe was 43,329, Ketu South was 39,119 while Ho West and Nkwanta South had 23,875 and 22,733 respectively. Based on these populations, Yamane's [26] formula for sample size determination was used to determine the final number of households. The formula is given as $n = \frac{N}{1+N(\alpha)^2}$ Where n is the sample size to be determined, N is the study population, and $\alpha \cong$ is the margin of error which was 0.05 at a significance level of 95%. Based on the formula, and adding 10% non-response rate, the minimum sample sizes for the four districts were 436 for Ketu South, 433 for Ho West, 437 for Hohoe, and 432 for Nkwanta South.

## Data collection

Three sets of questionnaires were used in the survey. These were 1) a household questionnaire which was used to collect data on all household members (usual residents), the household, and the dwelling, 2) a women's questionnaire administered in each household to all women aged 15–59 years and caregivers of children aged 13 years and younger; and 3) a men's questionnaire which was administered to all men aged 15–59 years in the households. The questionnaires were pretested in the Ho Municipality (Klefe) among 287 households. The pretesting allowed for further modification, of the framing and sequence of questions and as well determined the time for each questionnaire to be completed.

Data collection with the instruments was done in the form of computer-assisted personal interviews (CAPI). The digitised instruments (CAPI) were installed on the smartphones of the data collectors who were students of UHAS. The data collectors were trained for three days to acquaint them with the purpose of the study and how to administer the CAPI. There were about 25 data collectors in each district. Data collection activities were strictly supervised by experienced researchers from UHAS.

## Study variables

**Outcome variables.** The outcome variables of the study were depression, anxiety, and stress which were measured using the DASS-21 scale [27]. DASS-21 scale is a 21-time set of three self-report scales designed to measure emotional states of depression, anxiety and stress. Subscales of depression, anxiety and stress were made of seven (7) questions each. The depression subscale assesses dysphoria, hopelessness, devaluation of life, self-deprecation, lack of interest, anhedonia and inertia. The anxiety subscale measures autonomic arousal, skeletal muscle effects, situational anxiety, and subjective experience and anxious affect. The stress subscale measure relaxation difficulty, nervous arousal, being easily upset and over-reaction and impatient. The items were rated on a 4-point Likert Scale comprising did not apply to me at all (0), applied to me to some degree (1), applied to me to a considerable degree (2) and applied to me very much (3). Subscale scores were multiplied by 2 to obtain the final score of depression, anxiety and stress [27]. The validity and reliability of the DASS-21 scale is adequately established in other studies [28–31]. The DASS-21 scale in our study also exhibited a strong reliability coefficient (Cronbach's $\alpha$ = 0.925).

**Explanatory variables.** District of residence, age, sex, ethnicity, religion, educational status, occupation, income, current smoking status, current alcohol intake status, and hypertensive status were the explanatory variables. S2 Table is an appendix of all the study variables. Hypertension was classified based on the recommended cut-offs [32] as follows: Normal-(Systolic BP <120 and Diastolic BP <80 mmHg) and Hypertension, thus Pre-hypertension (Systolic BP = 120–139 and/or Diastolic BP = 80–89 mmHg); Hypertension- Stage I hypertension (Systolic BP = 140–159 and/or Diastolic BP = 90–99 mmHg) and Stage II hypertension (Systolic BP > 160 and/or Diastolic BP > 100 mmHg).

## Statistical analysis

Data collected were analysed using Stata 16/MP (StataCorp LP, Texas, USA). Firstly, means and standard deviations were used to describe continuous variables whereas proportion was used to describe categorical variables. District, religion, occupation, education, age, income, smoking, systolic and diastolic variables had missing data and were not included in the analysis. The prevalence of depression, anxiety and stress were estimated using the reference provided by Lovibond [27]. Depression, anxiety and stress were further classified into a dichotomous variable [33,34]. Overall, proportion of mental health conditions or psychological distress was ascertained by a respondent testing positive for at least depression, anxiety, or stress. Four comorbidities were established, namely: having depression and anxiety only, depression and stress only, anxiety and stress only, and having all the three (depression, anxiety, and stress).

Secondly, bivariate associations between depression, anxiety, stress and continuous variables (age, income, systolic and diastolic blood pressure measurements) were conducted using Pearson's correlation coefficients. To determine the predictors of depression, anxiety, and stress, we employed two binary logistic regression models (univariable and multivariable). Variables that showed significance (p<0.2) in the univariable analysis were included in the multivariable model in which statistical significance was considered at p<0.05.

## Ethical issues

Ethical clearance for the survey was obtained from the UHAS Review Ethics Committee (UHAS-REC A.6 [7] 17/18). Permission was also sought from the district/municipal health directorates and traditional authorities of the various communities before data were collected. Informed consent was obtained from participants before including them in the study. This

was achieved by giving them informed consent forms to sign/thumbprint. Confidentiality was ensured by using pseudonyms instead of the real names and other characteristics that personally identified the participants.

## Results

### Background characteristics of participants

Table 1 presents the background characteristics of the study participants. Out of the 2456 adults included in this study, females constituted 56.3%. The mean age was 41.8±0.3 and the comparative majority (24.8%) were 18 to 29 years old. Most of the participants were Ewes (80.3%) and Christians (87.4%). In terms of formal education, 40.2% had Junior High School (JHS) level of education. The comparative majority (30.6%) were into agriculture and most of them (70%) earned GH₵500–999 monthly. The prevalence of hypertension was 31%. Participants who were smokers at the time of data collection constituted 3.6% and 48% consumed alcohol.

### Prevalence of depression, anxiety, and stress

Fig 1 presents the prevalence of depression, anxiety, and stress. Overall, 51.8% of the participants experienced either depression, anxiety or stress. Participants constituting 53.3%, 25.2% and 9.7% experienced anxiety, depression and stress respectively. Concurrently, 8.4% experienced depression and stress alone, 9.6% had stress and anxiety alone, and 24.2% experienced depression and anxiety alone. Participants constituting 8.3% experienced all three mental health issues concurrently.

### Correlation between depression, anxiety and stress and continuous variables

Table 2 presents the correlation between continuous variables and the outcome variables (depression, anxiety, and stress). Systolic blood pressure (BP) had a positive correlation with anxiety. There was also a significant negative correlation between depression, anxiety, stress and diastolic BP. Age had a significant positive correlation with anxiety and stress.

### Predictors of depression, anxiety, and stress

Table 3 presents the predictors of depression, anxiety, and stress. Level of education and income significantly predicted all three outcome variables in our analysis. Adults with some formal education, for instance, had lower odds of experiencing depression, anxiety, and stress than those with no formal education. Those with a tertiary level of education were 68% (AOR = 0.32, 95%CI = 0.15–0.66), 65% (AOR = 0.35, 95%CI = 0.17–0.73), and 50% (AOR = 0.50, 95%CI = 0.33–0.76) less likely to experience depression, anxiety, and stress, respectively compared with those who had no formal education. Regarding income, participants in the highest wealth quintile were respectively less likely to experience depression (AOR = 0.27, 95%CI = 0.11–0.66), anxiety (AOR = 0.34, 95%CI = 0.13–0.86), and stress (AOR = 0.38, 95%CI = 0.16–0.90) compared with those who earned no income. Ethnicity and occupation significantly predicted depression and stress. Guans, for instance, had the lowest probabilities of experiencing depression (AOR = 0.40, 95%CI = 0.23–0.68) and stress (AOR = 0.33, 95%CI = 0.16–0.67). There was, however, no discernible pattern regarding the probability of experiencing depression and stress by occupation. Sex significantly predicted only stress while alcohol consumption predicted only anxiety. Females were, for instance, 1.48 times (95%CI = 1.12–1.95 p = 0.006) more likely to report being stressed than males while

**Table 1.** Background characteristics of study participants.

| Variable | Men | Women | Total |
|---|---|---|---|
| | [N = 1073] | [N = 1383] | [N = 2456] |
| **Study district** | | | |
| Ketu South | 317 (29.8) | 507 (37.0) | 824 (33.8) |
| Ho West | 288 (27.0) | 359 (26.2) | 647 (26.6) |
| Hohoe | 241 (22.6) | 290 (21.2) | 531 (21.8) |
| Nkwanta South | 219 (20.6) | 213 (15.6) | 432 (17.8) |
| **Mean age (±SD)** | 43.4±0.5 | 40.6±0.4 | 41.8±0.3 |
| **Age (In years)** | | | |
| 18–29 | 262 (24.7) | 336 (24.9) | 598 (24.8) |
| 30–39 | 213 (20.1) | 378 (28.0) | 591 (24.5) |
| 40–49 | 224 (21.2) | 271 (20.1) | 495 (20.5) |
| 50–59 | 178 (16.8) | 200 (14.8) | 378 (15.7) |
| 60+ | 182 (17.2) | 165 (12.2) | 347 (14.4) |
| **Ethnicity** | | | |
| Ewe | 788 (78.0) | 1060 (82.0) | 1848 (80.3) |
| Akan | 19 (1.9) | 27 (2.1) | 46 (2.0) |
| Guan | 104 (10.3) | 107 (8.3) | 211 (9.2) |
| Mole-Dagbani | 77 (7.6) | 63 (4.9) | 140 (6.1) |
| Others | 22 (2.2) | 35 (2.7) | 57 (2.5) |
| **Religion** | | | |
| Christianity | 903 (85.8) | 1204 (88.6) | 2107 (87.4) |
| Muslim | 36 (3.4) | 44 (3.2) | 80 (3.3) |
| Traditionalist | 113 (10.7) | 111 (8.2) | 224 (9.3) |
| **Educational status** | | | |
| None | 97 (9.2) | 246 (18.3) | 343 (14.3) |
| Primary | 112 (10.6) | 246 (18.3) | 358 (15.0) |
| JHS | 394 (37.5) | 567 (42.3) | 961 (40.2) |
| SHS | 299 (28.4) | 208 (15.5) | 507 (21.2) |
| Tertiary | 150 (14.3) | 74 (5.5) | 224 (9.4) |
| **Occupation** | | | |
| Unemployed | 81 (7.6) | 161 (12.2) | 242 (10.1) |
| Agriculture | 427 (40.1) | 302 (22.8) | 729 (30.6) |
| Sales and Service | 101 (9.5) | 573 (43.3) | 674 (28.2) |
| Skilled Manual | 231 (21.7) | 195 (14.8) | 426 (17.9) |
| Unskilled Manual | 104 (9.8) | 24 (1.8) | 128 (5.4) |
| Professional/Technical | 94 (8.8) | 50 (3.8) | 144 (6.0) |
| Others | 26 (2.4) | 17 (1.3) | 43 (1.8) |
| **Income (GH₵)** | | | |
| <500 | 19 (2.4) | 23 (2.4) | 42 (2.4) |
| 500–999 | 477 (59.4) | 748 (78.9) | 1225 (70.0) |
| ≥1000 | 172 (21.4) | 97 (10.2) | 269 (15.4) |
| **Hypertensive status** | | | |
| Normal | 454 (68.3) | 636 (69.5) | 1090 (69.0) |
| Hypertensive | 211 (31.7) | 279 (30.5) | 490 (31.0) |
| **Current smoking status** | | | |
| Non-current smoker | 966 (92.5) | 1342 (99.4) | 2308 (96.4) |
| Current smoker | 78 (7.5) | 8 (0.6) | 86 (3.6) |

(*Continued*)

**Table 1.** (Continued)

| Variable | Men | Women | Total |
|---|---|---|---|
|  | [N = 1073] | [N = 1383] | [N = 2456] |
| **Current alcohol intake status** |  |  |  |
| Non-current drinker | 362 (34.0) | 901 (66.0) | 1263 (52.0) |
| Current drinker | 703 (66.0) | 465 (34.0) | 1168 (48.0) |

adults who drank alcohol had a higher probability of experiencing anxiety (AOR = 1.52, 95% CI = 1.16–1.98).

## Discussion

This was a community-based cross-sectional study of the prevalence and predictors depression, anxiety, and stress among 2456 adults using data from the UHAS-Yonsei University

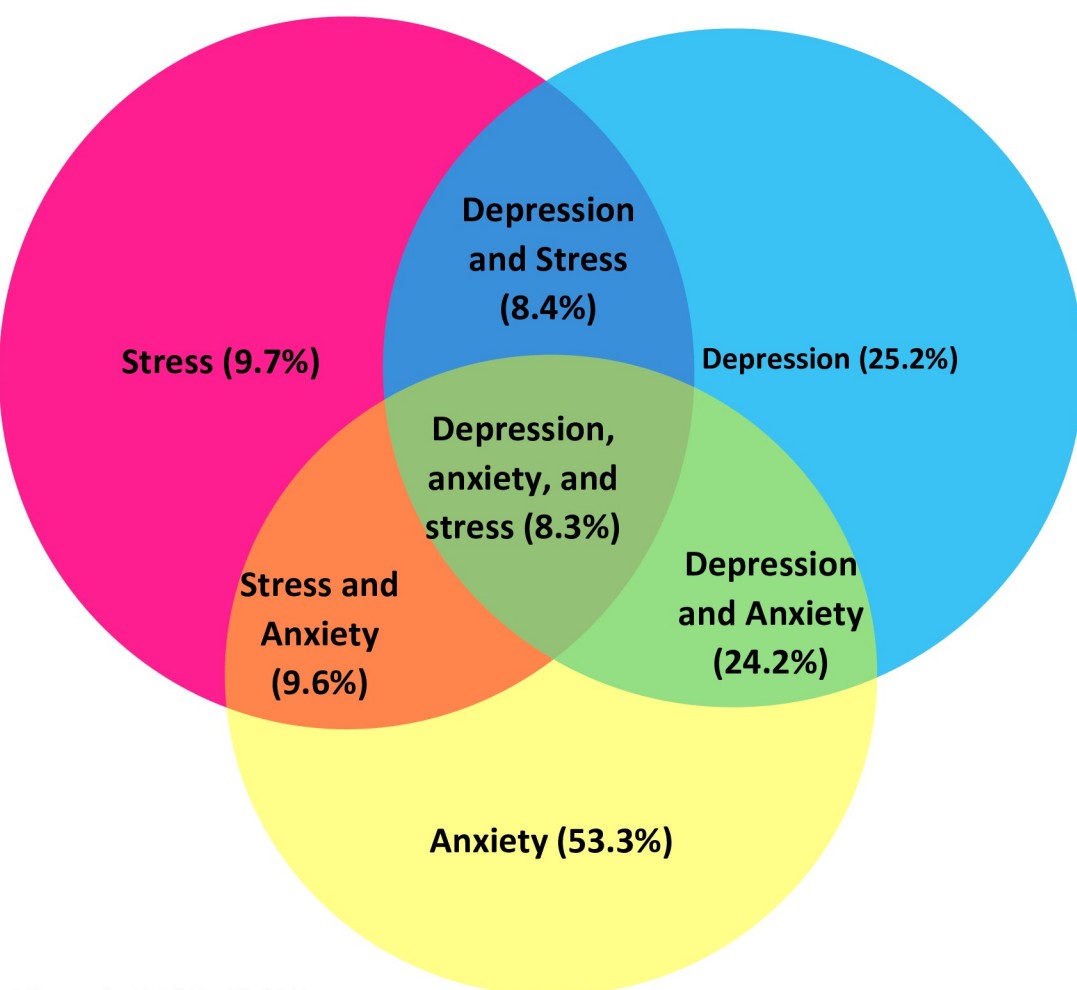

- Normal: (1185: 48.2%)
- Mental health condition (1273: 51.8%)

**Fig 1. Prevalence of depression, anxiety and stress.**

**Table 2. Correlations, means, standard deviations and Cronbach's α between background characteristics and depression, anxiety and stress.**

| Parameter | 1 | 2 | 3 | 4 | 5 | 6 | 7 | 8 |
|---|---|---|---|---|---|---|---|---|
| 1-Depression | 1 | | | | | | | |
| 2-Anxiety | 0.8314* | 1 | | | | | | |
| 3-Stress | 0.8303* | 0.8310* | 1 | | | | | |
| 4-Overall DASS-21 | 0.9414* | 0.9391* | 0.9403* | 1 | | | | |
| 5-Age | 0.022 | 0.0449* | 0.0405* | 0.0382 | 1 | | | |
| 6-Income | -0.0214 | -0.0149 | 0.0065 | -0.0135 | -0.0241 | 1 | | |
| 7-Systolic BP | 0.0229 | 0.0545* | 0.032 | 0.0367 | 0.0012 | 0.0301 | 1 | |
| 8-Diastolic BP | -0.0568* | -0.0588* | -0.0591* | -0.0620* | -0.0099 | -0.0112 | -0.0063 | 1 |
| Min-Max scores | 0–28 | 0–28 | 0–28 | 0–84 | 18–110 | 10–1000 | 53–235 | 33–183 |
| Mean (±SD) | 10.7 ± 5.0 | 10.9 ± 4.9 | 11.3 ± 5.1 | 32.8 ± 14.1 | 41.8 ± 15.5 | 419.0 ± 602 | 126.4 ± 21.4 | 79.3 ± 14.3 |
| n (Cronbach's α) | 7 (0.87) | 7 (0.83) | 7 (0.85) | 21 (0.94) | - | - | - | - |

Note: ***p<0.001.

n-number of items measured.

α-alpha.

BP-Blood Pressure measurement.

Partnership Project. Overall, 51.8% of the participants had at least a psychological distress: depression, anxiety, or stress. The prevalence of anxiety in our study was higher than those observed by Kusi-Mensah et al. (7.25%) [35], and Maideen et al. (8.2%) elsewhere [36]. The prevalence of depression observed in this study was also higher than those reported by previous studies including 11.9% [37], 6.7% [38], and 16.8% [39] in Ghana. This difference may be attributed the differences in socio-cultural and economic settings as well assessment tools used by the different studies. The prevalence of comorbidities of depression, anxiety, and stress found in the present study have negative implications for attainment of the SDG 3.4 target of promoting mental health and well-being for all by the year 2030 [1]. The high prevalence also highlights the burden of poor mental health among the populace at the backdrop that Ghana like most developing countries have limited available and accessible mental healthcare services. Consequently, initial symptoms may be undetected and thus have exacerbated among majority of the population. The poor mental health service availability has been captured to be resolved by the promulgation of the Mental Health Act in 2012, making provision for the establishment of a mental health fund which can be used to increase access to preventive and management services [13].

In our study, educational status and income significantly predicted the prevalence of depression, anxiety, and stress. Higher educational achievement, for instance, served as an important protective factor against mental health issues. This observation is congruent to previous postulations that a higher educational achievement has protective effects against mental health problems and this protection increases as one progress on the educational ladder [40–44]. The finding regarding education could be explained that a more educated individual is better informed about the symptoms/risk factors of mental health issues and, thus, institutes measures to prevent or deal with them appropriately when they occur [44,45]. Also, a more educated person has high likelihoods of positively constructing and embedding cohesive social structures [44,45].

We found that a high income serves as a protective factor that reduces the probability of experiencing depression, anxiety, and stress. This finding relates to a study by Patel et al. in which the authors posited that income inequality increases the risk of psychological distress

**Table 3. Predictors of depression, anxiety and stress.**

| Parameters | Depression | | Anxiety | | Stress | |
|---|---|---|---|---|---|---|
| | COR (95% CI) p-Value | AOR (95% CI) p-Value | COR (95% CI) p-Value | AOR (95% CI) p-Value | COR (95% CI) p-Value | AOR (95% CI) p-Value |
| **Sex** | | | | | | |
| Male | Ref. | | Ref. | | Ref | Ref |
| Female | 1.09 (0.90, 1.31) 0.375 | | 1.05 (0.89–1.23) 0.527 | | 1.39 (0.88, 2.20) 0.158 | 1.48 (1.12, 1.95) **0.006** |
| **Age (In years)** | | | | | | |
| 18–29 | Ref | Ref. | Ref | Ref. | Ref | |
| 30–39 | 1.28 (0.98, 1.66) 0.072 | 1.31 (0.90, 1.90) 0.160 | 0.96 (0.77, 1.21) 0.754 | 1.06 (0.72, 1.57) 0.755 | 1.06 (0.72, 1.54) 0.779 | |
| 40–49 | 1.30 (0.98, 1.71) 0.065 | 1.40 (0.95, 2.05) 0.089 | 1.21 (0.95, 1.54) 0.119 | 1.28 (0.85, 1.92) 0.242 | 0.91 (0.60, 1.37) 0.647 | |
| 50–59 | 1.21 (0.89, 1.63) 0.220 | 1.22 (0.80, 1.86) 0.356 | 1.14 (0.88, 1.47) 0.331 | 1.30 (0.83, 2.02) 0.255 | 1.16 (0.76, 1.77) 0.479 | |
| 60+ | 1.13 (0.83, 1.55) 0.428 | 1.03 (0.64, 1.66) 0.888 | 1.15 (0.88, 1.50) 0.300 | 0.94 (0.58, 1.55) 0.819 | 0.85 (0.53, 1.35) 0.492 | |
| **Ethnicity** | | | | | | |
| Ewe | Ref | Re | Ref | Ref | Ref | Ref |
| Akan | 0.77 (0.38, 1.56) 0.464 | 1.02 (0.46, 2.25) 0.956 | 1.05 (0.58, 1.90) 0.864 | 1.84 (0.73, 4.63) 0.197 | 1.06 (0.31, 3.58) 0.930 | 0.58 (0.18, 1.88) 0.363 |
| Guan | 0.55 (0.38, 0.80) 0.002 | 0.40 (0.23, 0.68) **0.001** | 0.63 (0.47, 0.83) 0.001 | 1.02 (0.63, 1.66) 0.944 | 0.34 (0.12, 0.94) 0.038 | 0.33 (0.16, 0.67) **0.002** |
| Mole-Dagbani | 0.82 (0.54, 1.23) 0.337 | 0.73 (0.41, 1.30) 0.285 | 0.76 (0.54, 1.08) 0.127 | 0.66 (0.34, 1.28) 0.215 | 0.77 (0.30, 1.93) 0.572 | 0.57 (0.29, 1.14) 0.110 |
| Others | 0.74 (0.39, 1.40) 0.354 | 0.55 (0.21, 1.41) 0.212 | 0.97 (0.57, 1.64) 0.897 | 0.95 (0.42, 2.16) 0.902 | 0.28 (0.04, 2.18) 0.226 | 0.46 (0.14, 1.49) 0.194 |
| **Religion** | | | | | | |
| Christianity | Ref | Ref. | Ref | Ref | Ref | Ref. |
| Muslim | 0.78 (0.45, 1.36) 0.374 | 1.26 (0.58, 2.74) 0.558 | 0.93 (0.59, 1.45) 0.744 | 0.72 (0.33, 1.62) 0.432 | 1.21 (0.34, 4.29) 0.768 | 0.51 (0.18, 1.41) 0.194 |
| Traditionalist | 1.56 (1.16, 2.10) 0.003 | 1.32 (0.87, 1.99) 0.189 | 1.22 (0.92, 1.61) 0.160 | 1.34 (0.78, 2.30) 0.283 | 1.55 (0.87, 2.78) 0.139 | 1.44 (0.95, 2.19) 0.085 |
| **Educational status** | | | | | | |
| None | Ref | Ref | Ref | Ref | Ref | Ref |
| Primary | 0.60 (0.43, 0.84) 0.003 | 0.38 (0.24, 0.61) **<0.001** | 0.72 (0.53, 0.98) 0.034 | 0.49 (0.29, 0.84) **0.009** | 0.93 (0.67–1.29) 0.681 | 0.80 (0.57–1.11) 0.190 |
| JHS | 0.68 (0.52, 0.88) 0.004 | 0.50 (0.34, 0.74) **0.001** | 0.74 (0.58, 0.95) 0.020 | 0.56 (0.35, 0.91) **0.019** | 0.85 (0.64–1.11) 0.225 | 0.70 (0.53–0.94) **0.018** |
| SHS | 0.58 (0.43, 0.79) <0.001 | 0.43 (0.28, 0.68) **<0.001** | 0.58 (0.44, 0.76) <0.001 | 0.42 (0.25, 0.71) **0.001** | 0.68 (0.51–0.94) 0.020 | 0.58 (0.42–0.80) **0.001** |
| Tertiary | 0.41 (0.27, 0.61) <0.001 | 0.32 (0.15, 0.66) **0.002** | 0.52 (0.37, 0.73) <0.001 | 0.35 (0.17, 0.73) **0.005** | 0.59 (0.39–0.88) 0.011 | 0.50 (0.33–0.76) **0.001** |
| **Occupation** | | | | | | |
| Unemployed | Ref | Ref. | Ref | Ref. | Ref | Ref. |
| Agriculture | 1.15 (0.81, 1.63) 0.426 | 2.93 (1.23, 6.97) **0.015** | 0.92 (0.69, 1.23) 0.563 | 1.10 (0.54, 2.24) 0.799 | 1.10 (0.40, 3.04) 0.853 | 0.86 (0.51, 1.45) 0.570 |
| Sales and Service | 1.42 (1.00, 2.01) 0.048 | 3.68 (1.55, 8.71) **0.003** | 1.13 (0.84, 1.51) 0.427 | 1.78 (0.88, 3.61) 0.107 | 2.21 (0.82, 5.91) 0.115 | 1.62 (0.98, 2.67) 0.058 |
| Skilled Manual | 1.27 (0.88, 1.85) 0.207 | 3.33 (1.40, 7.94) **0.007** | 1.04 (0.76, 1.43) 0.808 | 1.67 (0.82, 3.40) 0.158 | 1.49 (0.54, 4.11) 0.446 | 1.21 (0.70, 2.09) 0.489 |
| Unskilled Manual | 1.19 (0.72, 1.97) 0.501 | 3.18 (1.21, 8.35) **0.019** | 1.01 (0.66, 1.55) 0.973 | 0.87 (0.38, 2.02) 0.749 | 1.55 (0.45, 5.30) 0.488 | 1.09 (0.52, 2.29) 0.823 |
| Professional/Technical/ managerial | 0.68 (0.40, 1.16) 0.158 | 2.83 (0.97, 8.27) 0.058 | 0.73 (0.48, 1.10) 0.134 | 1.60 (0.66, 3.90) 0.302 | 0.42 (0.09, 1.90) 0.259 | 0.30 (0.10, 0.89) **0.031** |
| Others | 0.82 (0.36, 1.86) 0.628 | 2.40 (0.64, 8.98) 0.192 | 0.75 (0.39, 1.44) 0.384 | 1.06 (0.33, 3.37) 0.924 | 1.49 (0.33, 6.66) 0.603 | 1.38 (0.49, 3.90) 0.537 |
| **Income (In GHAAAA₵)** | | | | | | |
| None | Ref. | Ref. | Ref. | Ref. | Ref | Ref. |
| <500 | 0.46 (0.24, 0.86) 0.015 | 0.24 (0.11, 0.54) **<0.001** | 0.56 (0.29, 1.08) 0.082 | 0.39 (0.17, 0.91) **0.030** | 0.22 (0.09, 0.54) 0.001 | 0.32 (0.15, 0.69) **0.004** |
| 500–999 | 0.46 (0.23, 0.90) 0.024 | 0.24 (0.10, 0.56) **0.001** | 0.57 (0.29, 1.13) 0.105 | 0.33 (0.14, 0.82) **0.017** | 0.23 (0.08, 0.63) 0.004 | 0.36 (0.15, 0.84) **0.018** |
| ≥1000 | 0.42 (0.21, 0.85) 0.015 | 0.27 (0.11, 0.66) **0.004** | 0.51 (0.26, 1.03) 0.061 | 0.34 (0.13, 0.86) **0.023** | 0.33 (0.12, 0.95) 0.039 | 0.38 (0.16, 0.90) **0.027** |
| **Hypertensive status** | | | | | | |
| Normal | Ref. | | Ref. | Ref. | Ref. | |
| Hypertensive | 0.99 (0.77,1.27) 0.958 | | 1.16 (0.94, 1.44) 0.169 | 1.01 (0.76, 1.36) 0.936 | 0.93 (0.65,1.32) 0.691 | |
| **Current Smoking status** | | | | | | |
| Non-current smoker | Ref. | | Ref. | | Ref. | |
| Current smoker | 1.22 (0.76, 1.96) 0.407 | | 1.26 (0.81, 1.96) 0.288 | | 1.08 (0.53, 2.18) 0.828 | |
| **Current alcohol intake status** | | | | | | |
| Non-current drinker | Ref. | Ref. | Ref | Ref. | Ref | Ref. |

*(Continued)*

**Table 3.** (Continued)

| Parameters | Depression | | Anxiety | | Stress | |
|---|---|---|---|---|---|---|
| | COR (95% CI) p-Value | AOR (95% CI) p-Value | COR (95% CI) p-Value | AOR (95% CI) p-Value | COR (95% CI) p-Value | AOR (95% CI) p-Value |
| Current drinker | 1.16 (0.97, 1.39) 0.112 | 1.22 (0.95, 1.56) 0.118 | 1.25 (1.07, 1.47) 0.006 | 1.52 (1.16, 1.98) **0.002** | 1.12 (0.76, 1.65) 0.580 | 1.24 (0.95, 1.63) 0.112 |

Note: Bold values signify statistical significance of p-values <0.05 at the multivariable level.

COR: Crude odds ratio.

AOR: Adjusted odds ratio.

[46]. The finding regarding income could be explained by the fact those with higher income have more financial strength to afford their daily needs and are also more likely to seek mental health care promptly whenever they experience any symptoms of mental health challenges [46].

In the present study, ethnicity significantly predicted the prevalence of depression and stress. This finding is consistent with findings of other studies which identified ethnicity as a major factor that greatly influences depression and stress [47,48]. This finding could be explained by the differences in cultural practices such as diet and beliefs as espoused in a previous study [49]. The finding also implies that government policy and interventions to mitigate mental health challenges need to be implemented with ethnic diversity considerations. The occupation of participants also statistically predicted depression and stress. This finding is consistent with previous literature [50–53] and could be due to the challenges such as low motivation, marginalization, low self-esteem, physical/emotional exhaustion emanating from a heavy workload, and long working periods experienced by people at their workplaces, as these are known risk factors of mental health problems [54].

We found that drinking alcohol significantly increases the risk of developing anxiety. This finding is consistent with the finding by Smith and Randall which found a significant association between alcoholism and anxiety [55]. This finding could be that individuals who become alcohol dependent mostly face financial and social challenges such as rejection and stigmatization which consequently makes them anxious [56,57]. Also, this could be because people who are alcoholic have higher tendencies of living with chronic non-communicable diseases like hypertension which is known to exert grievous strains on the mental health of victims [58–60].

We observed that gender is an important predictor of stress among the adult population. Our finding where females had a higher probability of being stressed is congruent to other studies which noted that females are more likely than males to report higher stress levels [61–63]. Our observation could be attributed to the fact that women in developing countries have more stressful household responsibilities including caring for children, carrying-out house chores and also, providing financial resources to support the household [64–67].

## Strengths and limitations

In this study, we demonstrated the presence of comorbidities in depression, anxiety and stress among adults in Ghana. This made it possible to exhaustively appreciate the magnitude of depression, anxiety, and stress as experienced by the adult Ghanaian population. Our use of correlation and logistic regression also ensured that we robustly established the relationships that exist between our outcome and explanatory variables. A limitation of our study, however, is the fact that we depended on verbal reports of the adult community members. The responses

given by the study participants might have, therefore, suffered response and recall biases. Our study also reported missing values which could affect the findings we made.

## Conclusion

Depression, anxiety, and stress are fairly high in this study. There was also quite a high comorbidity of the three mental health issues among the adult population. These findings preclude Ghana's ability to achieve the SDG 3.4 target of promoting mental health and well-being for all by the year 2030. To accelerate progress towards the achievement of the SDG target, there is a need to strengthen existing policies seeking to improve the mental health of the Ghanaian populace. Specific strategies could include aggressive implementation of the country's 2012 Mental Health Act which makes provisions for the establishment of a Mental Health Fund. This would improve the financial circumstances of indigenes as income has been realised in the present study as an important factor influencing depression, anxiety, and stress among the adult population.

## Supporting information

**S1 Table. STROBE checklist.**
(DOC)

**S2 Table. Study variables.**
(DOCX)

## Acknowledgments

### Declaration

We than the UHAS-Yonsei Project Group for their support throughout the study.

## Author Contributions

**Conceptualization:** Hubert Amu, Eric Osei, Philip Kofie, Richard Owusu, Kennedy Diema Konlan, Eunji Kim, Verner Ndudiri Orish, Emmanuel Manu, Phyllis Atta Parbey, Prince Kubi Appiah, Joyce Komesuor, Elvis Enowbeyang Tarkang, Robert Kaba Alhassan, Ernestina Safoa Donkor, Francis Bruno Zottor, Margaret Kweku, Paul Amuna, So Yoo Kim.

**Data curation:** Hubert Amu, Philip Kofie, Samuel Adolf Bosoka, Martin Adjuik.

**Formal analysis:** Hubert Amu, Philip Kofie, Samuel Adolf Bosoka, Martin Adjuik, Margaret Kweku.

**Funding acquisition:** Eunji Kim, Prince Kubi Appiah, Robert Kaba Alhassan, Ernestina Safoa Donkor, Margaret Kweku, John Owusu Gyapong.

**Investigation:** Hubert Amu, Philip Kofie, Richard Owusu, Samuel Adolf Bosoka, Kennedy Diema Konlan, Eunji Kim, Verner Ndudiri Orish, Emmanuel Manu, Phyllis Atta Parbey, Farrukh Ishaque Saah, Hadiru Mumuni, Prince Kubi Appiah, Joyce Komesuor, Martin Amogre Ayanore, Gregory Kofi Amenuvegbe, Martin Adjuik, Elvis Enowbeyang Tarkang, Ernestina Safoa Donkor, Margaret Kweku, So Yoo Kim.

**Methodology:** Hubert Amu, Eric Osei, Philip Kofie, Richard Owusu, Samuel Adolf Bosoka, Kennedy Diema Konlan, Eunji Kim, Verner Ndudiri Orish, Raymond Saa-Eru Maalman, Emmanuel Manu, Phyllis Atta Parbey, Farrukh Ishaque Saah, Hadiru Mumuni, Joyce

Komesuor, Martin Amogre Ayanore, Gregory Kofi Amenuvegbe, Siwoo Kim, Hajun Jung, Martin Adjuik, Elvis Enowbeyang Tarkang, Ernestina Safoa Donkor, Francis Bruno Zottor, Margaret Kweku, Paul Amuna, So Yoo Kim.

**Project administration:** Hubert Amu, Eric Osei, Philip Kofie, Richard Owusu, Samuel Adolf Bosoka, Kennedy Diema Konlan, Eunji Kim, Verner Ndudiri Orish, Raymond Saa-Eru Maalman, Emmanuel Manu, Phyllis Atta Parbey, Farrukh Ishaque Saah, Hadiru Mumuni, Prince Kubi Appiah, Joyce Komesuor, Martin Amogre Ayanore, Gregory Kofi Amenuvegbe, Siwoo Kim, Hajun Jung, Martin Adjuik, Elvis Enowbeyang Tarkang, Robert Kaba Alhassan, Ernestina Safoa Donkor, Francis Bruno Zottor, Margaret Kweku, Paul Amuna, So Yoo Kim, John Owusu Gyapong.

**Resources:** Hubert Amu, Kennedy Diema Konlan, Eunji Kim, Verner Ndudiri Orish, Emmanuel Manu, Hadiru Mumuni, Prince Kubi Appiah, Robert Kaba Alhassan, Ernestina Safoa Donkor, Margaret Kweku, Paul Amuna, So Yoo Kim, John Owusu Gyapong.

**Software:** Samuel Adolf Bosoka, Martin Adjuik.

**Supervision:** Hubert Amu, Kennedy Diema Konlan, Eunji Kim, Verner Ndudiri Orish, Raymond Saa-Eru Maalman, Emmanuel Manu, Hadiru Mumuni, Prince Kubi Appiah, Joyce Komesuor, Martin Amogre Ayanore, Gregory Kofi Amenuvegbe, Hajun Jung, Martin Adjuik, Elvis Enowbeyang Tarkang, Robert Kaba Alhassan, Ernestina Safoa Donkor, Francis Bruno Zottor, Margaret Kweku, Paul Amuna, So Yoo Kim, John Owusu Gyapong.

**Validation:** Hubert Amu, Eric Osei, Richard Owusu, Kennedy Diema Konlan, Eunji Kim, Raymond Saa-Eru Maalman, Phyllis Atta Parbey, Hadiru Mumuni, Martin Amogre Ayanore, Siwoo Kim, Martin Adjuik, Elvis Enowbeyang Tarkang, Ernestina Safoa Donkor, Francis Bruno Zottor, Margaret Kweku, Paul Amuna, So Yoo Kim, John Owusu Gyapong.

**Visualization:** Eric Osei, Samuel Adolf Bosoka, Verner Ndudiri Orish, Farrukh Ishaque Saah, Martin Amogre Ayanore, Martin Adjuik.

**Writing – original draft:** Hubert Amu, Philip Kofie, Samuel Adolf Bosoka, Phyllis Atta Parbey, Farrukh Ishaque Saah.

**Writing – review & editing:** Hubert Amu, Eric Osei, Philip Kofie, Richard Owusu, Samuel Adolf Bosoka, Kennedy Diema Konlan, Eunji Kim, Verner Ndudiri Orish, Raymond Saa-Eru Maalman, Emmanuel Manu, Phyllis Atta Parbey, Farrukh Ishaque Saah, Hadiru Mumuni, Prince Kubi Appiah, Joyce Komesuor, Martin Amogre Ayanore, Gregory Kofi Amenuvegbe, Siwoo Kim, Hajun Jung, Martin Adjuik, Elvis Enowbeyang Tarkang, Robert Kaba Alhassan, Ernestina Safoa Donkor, Francis Bruno Zottor, Margaret Kweku, Paul Amuna, So Yoo Kim, John Owusu Gyapong.

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
