## [Decision Letter · Decision Letter 0]

6 Jul 2021

PONE-D-21-06572

Prevalence and predictors of depression, anxiety, and stress among adults in Ghana: A community-based cross-sectional study

PLOS ONE

Dear Dr. Amu,

Thank you for submitting your manuscript to PLOS ONE. After careful consideration, we feel that it has merit but does not fully meet PLOS ONE’s publication criteria as it currently stands. Therefore, we invite you to submit a revised version of the manuscript that addresses the points raised during the review process.

The reviewers found merit in the findings of this study, particularly given the lack of mental health research in Ghana and the broader Sub-Saharan African region. The reviewers did, however, express some concerns with the current version of the manuscript, which need to be addressed in order for the paper to move forward in the review process. Importantly, you need to clarify key aspects of the methods including clarification about how the region and households for the study were selected and the contradictory statements about the inclusion/exclusion age for the study. In one section of the study, you suggest that parental consent was obtained, but in another section you mention that everyone in the study was above the age of 18 years. Clarity around this is needed. There is a need for the authors to provide a deeper literature review in the introduction and/or to provide a deeper discussion of how study findings differ from/corroborate findings from previous research/literature. A deeper discussion of why findings differ from/corroborate previous research/literature is also advised. Please see reviewers' detailed comments below.

We look forward to receiving your revised manuscript.

Kind regards,

Samantha C Winter, Ph.D.

Academic Editor

PLOS ONE

Journal Requirements:

2. Please include a copy of Table 3 which you refer to in your text on page 17.

3. We note you have included a table to which you do not refer in the text of your manuscript. Please ensure that you refer to Table 5 in your text; if accepted, production will need this reference to link the reader to the Table.

4. Please upload a copy of Supporting Information Table S1 which you refer to in your text on page 7.

Reviewers' comments:

Reviewer's Responses to Questions

**Comments to the Author**

1. Is the manuscript technically sound, and do the data support the conclusions?

Reviewer #1: Yes

Reviewer #2: No

2. Has the statistical analysis been performed appropriately and rigorously? 

Reviewer #1: Yes

Reviewer #2: Yes

3. Have the authors made all data underlying the findings in their manuscript fully available?

Reviewer #1: No

Reviewer #2: No

4. Is the manuscript presented in an intelligible fashion and written in standard English?

Reviewer #1: No

Reviewer #2: Yes

5. Review Comments to the Author

Reviewer #1: The authors have examined a neglected and increasing area of concern globally particularly during the COVID-19 pandemic. Considering the large sample size and the first study of this nature in the country, the findings could inform policy and practice if adequately disseminated. The authors have clearly described the details of each condition under study which provides an opportunity for wider readership including students and clinicians who are non-specialists in the field.

Introduction

Page 4: The use of “sufferers and suffer” for people with mental illness is stigmatizing. Words such as persons with/living with/experienced mental illness etc are preferred by experts with experience.

Methods section: Well described. However, it is not clear how the Volta region was selected. How were the households selected in the identified communities?

Page 8: For the current publication, however, only adults 18+ years old were included in the study, yet on page 12, Parental assent was obtained for participants below 18 years. Please clarify.

Results: The prevalence for this study was very high compared to other studies in the same region. What could be the reasons for this? Different culture or methods used? The authors could mention if the assessment tool is a screening or diagnostic? How were the tools administered?

The authors found that high income serves as a protective factor. This publication is written in the context of COVID-19 that has direct impact on economy/income. This manuscript could be further strengthened by illustrating the implication of the study in relation to the current situation. For instance - could provide opportunities (the authors can be specific) for programme development and/or amendments as recommended by the authors.

Page 22: The use of the term “sex” in the last paragraph of discussion could be replaced with gender due to the gender norms described at the end of the sentence.

Conclusion: Check for typos.

Reviewer #2: Thank you very much for the opportunity to review this interesting manuscript. Mental health related research in lacking generally in African and the authors attempt to fill a relevant gap in Ghana is commendable. Unfortunately, there are several conceptual and methodologies inadequacies in the current manuscript that should be addressed to warrant possible publication. This identified issues reduces the incremental contribution of this manuscript to scientific discourse in the area of mental in Ghana in particular and the sub-region in of this manuscript is very negligible. I have listed these issues below.

1. In the introduction section, the rationale on which the study was based could be strengthened. The authors literally mentioned on the face values the findings of these studies without providing much further details. For example, the authors stated that "Arhin et al. [15] studied anxiety, stress, and

depression and bullying victimization among school-going adolescents". But this is not enough, the authors should provide further information and the actual key findings of such studies that are reviewed in the introduction section. The study by Arhin et al. indicated that "Findings revealed that bullying victimisation was associated with elevated levels of depression, anxiety and stress; and that only depression was found as a significant predictor of bullying victimisation among school-going adolescents in Accra, Ghana". An expanded and detailed description gives the better picture of the situation and thus able to appreciate the scientific gap that the current study intends to fill.

2. Again, if the authors intend to make reference to the prevalence of depression, stress and anxiety in the discussion, then such information should be presented in the introduction to help the author make meaning from what is being discussed. For example, in the discussion it was reported that "The prevalence of depression observed in this study was also higher than those reported by previous studies [32–34]". It remains unclear what these studies are and what were the prevalent rates of depression as found in these 3 studies.

3. Furthermore, I also observed that almost all the explanatory variables used in the study were not adequately reviewed in the introduction section. A cursory look on studies in Ghana can provide adequate information as how each of these explanatory variables were included in the analysis and generation of results. I do not want to assume that the authors included these variables in the study without adequate review of literature. For example, In the study by Asante and Andoh-Arthur, there was not any gender differences in terms of depressive symptoms among university study but alcohol consumption, engagement in HIV risk behaviour among others were found to be related to depressive symptoms. At least 2 paragraph detailing how each of the explanatory variables were related to each of the outcome variables should be adequate. If there are no contextually relevant studies in Ghana on a key variable, the authors could reply on a study conducted with the sub-region.

4. The authors mentioned that "Overall, 51.8% of the participants experienced either depression, anxiety or stress" How did the authors obtained this figure and I also think that the strategy used in obtaining values in Fig 1 should be explained further in the methods section, under data analysis.

5. Finally, due to the lack of adequate literature review, the authors were not able to provided contextually relevant explanation for the key finding. For example, the authors stated that " The

high prevalence also points to the fact that interventions instituted in the country to promote the

mental health and well-being of the populace are probably not yielding the expected results and,

therefore, necessary amendments are required. it is not clear how the high prevalence of depression, anxiety and stress among the sample in this study could possibly be due to the failed mental health policy? This is farfetched and not appropriate. Are there any community-based factors within the study area that could possibly play a role in the high prevalence reported in the study.

6. Also some of the recommendation of the authors are not entirely based on the findings of the study. For example, "Our findings warrant the strengthening of interventions by the government of

Ghana in collaboration with trade and labour institutions to improve working conditions and

support the emotional and mental health of both public and private sector workers".

6. PLOS authors have the option to publish the peer review history of their article (what does this mean?). If published, this will include your full peer review and any attached files.

Reviewer #1: **Yes: **Christine Musyimi

Reviewer #2: No

---

## [Author Response · Author response to Decision Letter 0]

22 Jul 2021

Reviewer 1:

Introduction

1. Page 4: The use of “sufferers and suffer” for people with mental illness is stigmatizing. Words such as persons with/living with/experienced mental illness etc are preferred by experts with experience.

Response: This has been revised.

Methods section:

2. Well described. However, it is not clear how the Volta region was selected. How were the households selected in the identified communities?

Response: The Volta region (now divided into Volta and Oti regions) was the project site for the UHAS-Yonsei University Partnership project as explained on page 6 of the manuscript. How households were identified and selected has, however, been further explained on page 9 of the manuscript.

3. Page 8: For the current publication, however, only adults 18+ years old were included in the study, yet on page 12, Parental assent was obtained for participants below 18 years. Please clarify.

Response: The text on parental assent has been deleted from the revised manuscript.

Results:

4. The prevalence for this study was very high compared to other studies in the same region. What could be the reasons for this? Different culture or methods used? The authors could mention if the assessment tool is a screening or diagnostic? How were the tools administered?

Response: This have been revised as suggested on pages 11 (lines 227-232) and 21 (lines 360-362) of the manuscript.

5. The authors found that high income serves as a protective factor. This publication is written in the context of COVID-19 that has direct impact on economy/income. This manuscript could be further strengthened by illustrating the implication of the study in relation to the current situation. For instance - could provide opportunities (the authors can be specific) for programme development and/or amendments as recommended by the authors.

Response: Although we acknowledge the impact of COVID-19 on economy/income as stated by the reviewer, the study was conducted before the onset of the pandemic. 

6. Page 22: The use of the term “sex” in the last paragraph of discussion could be replaced with gender due to the gender norms described at the end of the sentence.

Response: “sex” has been replaced with “gender” as suggested.

Conclusion:

7. Check for typos.

Response: Typos and grammatical issues have been addressed.

Reviewer 2:

Thank you very much for the opportunity to review this interesting manuscript. Mental health related research in lacking generally in African and the authors attempt to fill a relevant gap in Ghana is commendable. Unfortunately, there are several conceptual and methodologies inadequacies in the current manuscript that should be addressed to warrant possible publication. This identified issues reduces the incremental contribution of this manuscript to scientific discourse in the area of mental in Ghana in particular and the sub-region in of this manuscript is very negligible. I have listed these issues below.

1. In the introduction section, the rationale on which the study was based could be strengthened. The authors literally mentioned on the face values the findings of these studies without providing much further details. For example, the authors stated that "Arhin et al. [15] studied anxiety, stress, and depression and bullying victimization among school-going adolescents". But this is not enough, the authors should provide further information and the actual key findings of such studies that are reviewed in the introduction section. The study by Arhin et al. indicated that "Findings revealed that bullying victimisation was associated with elevated levels of depression, anxiety and stress; and that only depression was found as a significant predictor of bullying victimisation among school-going adolescents in Accra, Ghana". An expanded and detailed description gives the better picture of the situation and thus able to appreciate the scientific gap that the current study intends to fill.

Response: The introduction section has been revised to incorporate the suggestions as stated.

2. Again, if the authors intend to make reference to the prevalence of depression, stress and anxiety in the discussion, then such information should be presented in the introduction to help the author make meaning from what is being discussed. For example, in the discussion it was reported that "The prevalence of depression observed in this study was also higher than those reported by previous studies [32–34]". It remains unclear what these studies are and what were the prevalent rates of depression as found in these 3 studies.

Response: This has been revised to include the specific prevalence observed in those studies (see page 21, lines 359-371).

3. Furthermore, I also observed that almost all the explanatory variables used in the study were not adequately reviewed in the introduction section. A cursory look on studies in Ghana can provide adequate information as how each of these explanatory variables were included in the analysis and generation of results. I do not want to assume that the authors included these variables in the study without adequate review of literature. For example, In the study by Asante and Andoh-Arthur, there was not any gender differences in terms of depressive symptoms among university study but alcohol consumption, engagement in HIV risk behaviour among others were found to be related to depressive symptoms. At least 2 paragraph detailing how each of the explanatory variables were related to each of the outcome variables should be adequate. If there are no contextually relevant studies in Ghana on a key variable, the authors could reply on a study conducted with the sub-region.

Response: Some of literature review on the explanatory variables have been included in the introduction section on pages 5-6 (Lines 97-119). 

4. The authors mentioned that "Overall, 51.8% of the participants experienced either depression, anxiety or stress" How did the authors obtained this figure and I also think that the strategy used in obtaining values in Fig 1 should be explained further in the methods section, under data analysis.

Response: How Fig 1 was achieved has been explained further on page 12 of the manuscript.

5. Finally, due to the lack of adequate literature review, the authors were not able to provided contextually relevant explanation for the key finding. For example, the authors stated that " The high prevalence also points to the fact that interventions instituted in the country to promote the mental health and well-being of the populace are probably not yielding the expected results and, therefore, necessary amendments are required. it is not clear how the high prevalence of depression, anxiety and stress among the sample in this study could possibly be due to the failed mental health policy? This is farfetched and not appropriate. Are there any community-based factors within the study area that could possibly play a role in the high prevalence reported in the study?

Response: This has been addressed and restructured on page 21 of the manuscript.

6. Also some of the recommendation of the authors are not entirely based on the findings of the study. For example, "Our findings warrant the strengthening of interventions by the government of Ghana in collaboration with trade and labour institutions to improve working conditions and support the emotional and mental health of both public and private sector workers"

Response: The recommendations have been revised.

---

## [Decision Letter · Decision Letter 1]

20 Sep 2021

Prevalence and predictors of depression, anxiety, and stress among adults in Ghana: A community-based cross-sectional study

PONE-D-21-06572R1

Dear Dr. Amu,

We’re pleased to inform you that your manuscript has been judged scientifically suitable for publication and will be formally accepted for publication once it meets all outstanding technical requirements. Please ensure that the manuscript is reviewed carefully for typographical, editorial, and grammatical mistakes. For example, there are several typographical errors, including missing or misused articles and conjunctions. Your proofing process will be the last opportunity you have to edit these mistakes/issues prior to publication.

Kind regards,

Samantha C Winter, Ph.D.

Academic Editor

PLOS ONE

Additional Editor Comments (optional):

Reviewers' comments:

Reviewer's Responses to Questions

**Comments to the Author**

1. If the authors have adequately addressed your comments raised in a previous round of review and you feel that this manuscript is now acceptable for publication, you may indicate that here to bypass the “Comments to the Author” section, enter your conflict of interest statement in the “Confidential to Editor” section, and submit your "Accept" recommendation.

Reviewer #1: All comments have been addressed

2. Is the manuscript technically sound, and do the data support the conclusions?

Reviewer #1: Yes

3. Has the statistical analysis been performed appropriately and rigorously? 

Reviewer #1: Yes

4. Have the authors made all data underlying the findings in their manuscript fully available?

Reviewer #1: Yes

5. Is the manuscript presented in an intelligible fashion and written in standard English?

Reviewer #1: Yes

6. Review Comments to the Author

Reviewer #1: The revised version of the manuscript looks great. While the authors acknowledge that the study was conducted before the onset of the pandemic, the findings are reported in the context of COVID-19 and recommendations/statement to address income as a protective factor could be useful for readers now and in future pandemics.

7. PLOS authors have the option to publish the peer review history of their article (what does this mean?). If published, this will include your full peer review and any attached files.

Reviewer #1: **Yes: **Christine Musyimi

---

## [Editor Report · Acceptance letter]

29 Sep 2021

PONE-D-21-06572R1 

Prevalence and predictors of depression, anxiety, and stress among adults in Ghana: A community-based cross-sectional study 

Dear Dr. Amu:

I'm pleased to inform you that your manuscript has been deemed suitable for publication in PLOS ONE. Congratulations! Your manuscript is now with our production department. 

Kind regards, 

on behalf of

Dr. Samantha C Winter 

Academic Editor

PLOS ONE